# The relationship between object-based spatial ability and virtual navigation performance

Tanya Garg[1], Pablo Fernández Velasco[2], Eva Zita Patai[1,3], Charlotte P. Malcolm[1], Victor Kovalets[1], Veronique D. Bohbot[4], Antoine Coutrot[5], Mary Hegarty[6], Michael Hornberger[7], Hugo J. Spiers[1]*

1 Department of Experimental Psychology, Division of Psychology and Language Sciences, Institute of Behavioural Neuroscience, University College London, London, United Kingdom, 2 Department of Philosophy, Trinity College Dublin, Dublin, Ireland, 3 Department of Psychology, School of Biological and Behavioural Sciences, Queen Mary University, London, United Kingdom, 4 Faculty of Medicine, Department of Psychiatry, Douglas Mental Health University Institute, McGill University, McGill, Canada, 5 LIRIS—CNRS —Université de Lyon, Lyon, France, 6 Department of Psychological and Brain Sciences, University of California Santa Barbara, Santa Barbara, California, United States of America, 7 Norwich Medical School, University of East Anglia, Norwich, United Kingdom

* h.spiers@ucl.ac.uk

**Data Availability Statement:** All relevant data are within the manuscript and its supporting information files, except data for some analyses on "Wayfinding and Path Integration Performance"

## Abstract

Spatial navigation is a multi-faceted behaviour drawing on many different aspects of cognition. Visuospatial abilities, such as mental rotation and visuospatial working memory, in particular, may be key factors. A range of tests have been developed to assess visuospatial processing and memory, but how such tests relate to navigation ability remains unclear. This understanding is important to advance tests of navigation for disease monitoring in various disorders (e.g., Alzheimer's disease) where spatial impairment is an early symptom. Here, we report the use of an established mobile gaming app, Sea Hero Quest (SHQ), as a measure of navigation ability in a sample of young, predominantly female university students (N = 78; 20; female = 74.3%; mean age = 20.33 years). We used three separate tests of navigation embedded in SHQ: wayfinding, path integration and spatial memory in a radial arm maze. In the same participants, we also collected measures of mental rotation (Mental Rotation Test), visuospatial processing (Design Organization Test) and visuospatial working memory (Digital Corsi). We found few strong correlations across our measures. Being good at wayfinding in a virtual navigation test does not mean an individual will also be good at path integration, have a superior memory in a radial arm maze, or rate themself as having a strong sense of direction. However, we observed that participants who were good in the wayfinding task of SHQ tended to perform well on the three visuospatial tasks examined here, and to also use a landmark strategy in the radial maze task. These findings help clarify the associations between different abilities involved in spatial navigation.

## 1. Introduction

Navigation is a fundamental skill that underlies exploration and survival. The ability to effectively navigate involves a host of capacities such as planning routes, reading maps, recognizing

reported in the Appendix, which will be available from past work via the following link: https://doi.org/10.1038/s41586-022-04486-7.

**Funding:** HJS and M Hornberger received funding from the Alzheimer's Research UK (https://www.alzheimersresearchuk.org/), Grant (ARUK-DT2016-1). The funders had no role in study design, data collection and analysis, decision to publish, or preparation of the manuscript.

**Competing interests:** The authors have declared that no competing interests exist.

landmarks and keeping track of direction. Deficits in these competencies may constitute an early marker of degenerative conditions such as Alzheimer's disease (AD); furthermore, spatial abilities are impaired in various neurological disorders or conditions, such as multiple sclerosis, vestibular syndromes and autism [1–4]. Furthermore, navigation impairments like disorientation can affect one's quality of life by causing distress and impediments in daily functioning [5,6]. Therefore, understanding individual differences in competencies and strategies underlying navigation can be useful in not only identifying risk factors of cognitive decline, but also in designing tools, interventions and environments to cater to unique navigational needs. More broadly, knowledge of the complex mechanisms and interactions involved in navigation will advance the domain of spatial cognition at large.

Creating a valid standardised test of navigation is not easy. This is because of the difficulties in achieving the required levels of environmental manipulation and experimental control in standard research settings, which are further compounded by the problem of testing large enough cohorts to account for the wide variations in performance. In recent years, virtual reality (VR) and the widespread touch-screen technology on tablet and mobile devices have offered new possibilities for testing. Our team capitalised on these possibilities by developing a set of tests for navigation ability in the form of the video game app Sea Hero Quest (SHQ) [7]. We have employed SHQ to test the navigation ability of 3.9 million people across the world [8,9]. SHQ has good test-retest reliability [10], and has been shown to be predictive of real-world navigational performance [11].

Studies using SHQ have revealed that gender differences in navigation ability for a country can be partially attributed to gender inequality [8]. They have also found that individuals are more adept at navigating environments that are topologically similar to those in which they were raised [12]. Such studies have also shown that performance is best for older participants who report sleeping 7 hours per night compared to those reporting more or less sleep [13]. It has also been used in the study of AD, for instance, to detect sub-optimal navigation performance in pre-clinical AD, to classify spatial impairments in healthy participants at a high risk of AD [14], and to detect those AD patients most prone to disorientation [15]. Finally, SHQ has also been employed to detect wayfinding and path integration (PI) deficits in patients with traumatic brain injury [16].

Navigation performance as assessed by SHQ depends on visuospatial abilities [17–19]. These abilities comprise a set of distinct skills involved in perceiving and manipulating objects in space [20–22]. One such skill is mental rotation, which involves transforming a mental representation of an object to predict how it would look from different angles [23]. Mental rotation facilitates navigation by enabling the formation of precise and flexible mental models of the environment [24], which can be useful to enhance spatial orientation and navigation performance. In addition to the Mental Rotation Test (MRT) [23], we employ the recent Design Organisation Test (DOT) [25] as a measure of visuospatial processing, a cognitive competency involved in navigation [26]. A classic test of visuospatial processing is the Block Design sub-test of the Weshler scales, which involves rearranging blocks that have various colour patterns on different sides with the aim of matching a target pattern [27]. Previous studies have indicated a strong correlation between this test and neuropsychological measures of visuospatial ability [28] and everyday visuospatial skills [29]. Notably, the DOT correlates highly with the Block Design sub-test, but does not involve fine motor skills like in the latter, and is substantially quicker to administer [25]. Finally, we also test visuospatial working memory (VSWM) using the forward digital Corsi Block Tapping Test (D-Corsi) as it is another cognitive construct that underlies navigation [30]. Corsi performance requires participants to tap spatially separated blocks in a given sequence [31]. The two forms of Corsi, forward and backward, have been thought to have different cognitive demands [32]. Specifically, it has been

suggested that the backward condition assesses executive function and visuospatial short-term memory, whereas the forward condition assesses only the latter [33,34]. Therefore, the forward Corsi test may serve as a more accurate measure of VSWM.

Navigational ability depends not only on visuospatial abilities, but also on personal preferences and strategies, which are usually assessed using questionnaires. The type of preferred navigational strategy, as measured by the Navigational Strategies Questionnaire [35], may determine one's navigational performance. Navigating using a map-based strategy, relative to a non-map-based strategy, for instance, requires a fine-grained and integrated representation of the environment [36]. Individuals with a preference for the aforementioned strategy are generally considered to be better at navigation tasks (35), and to show greater flexibility when navigating [37]. Additionally, perception of one's own spatial orientation in the environment may also influence navigation. In fact, an association between performance on tasks requiring participants to update their location in space as a result of self-motion and scores on the SBSOD has been found [38]. Overall, the existing literature suggests that visuospatial abilities and navigational preferences and strategies are independent but related factors that contribute to navigation ability [18,39,40]. Nevertheless, the heterogeneity in testing in existing studies means that the nature of these connections requires further study.

We test the navigation abilities of a predominantly female sample of university students on three tasks in SHQ: wayfinding, path integration (PI), and the radial arm maze (RAM) test of spatial memory. Wayfinding refers to goal-directed navigation to get from one place to another in the environment, while PI involves keeping track of one's location by using internal cues from one's movement [41]. The RAM tests both working memory and procedural memory processes [42,43]. Participants were also tested on three measures of visuospatial ability: the MRT Form A [23], the DOT [25], and the forward D-Corsi [31]. Further, participants also completed the Santa Barbara Sense of Direction Scale (SBSOD) [38], the Navigation Strategies Questionnaire (NSQ) [35], a multiple-choice question about their navigation strategy on RAM levels of SHQ and a questionnaire about perceived stress when navigating in SHQ.

In this study, we explore how visuospatial abilities, navigation strategy and gameplay stress relate to performance on SHQ and to each other. Based on the literature (e.g., Wolbers & Hegarty [17]), we made 20 predictions. We expected wayfinding to correlate with other measures due to its diverse demands such as perception, memory, decision-making, etc. Specifically, we predicted that longer duration to complete wayfinding levels (i.e., wayfinding inefficiency) would be associated with lower mental rotation, visuospatial processing, VSWM, sense of direction and mapping tendency as measured by the MRT, DOT, D-Corsi, SBSOD and NSQ, respectively. Greater wayfinding inefficiency would also be related to higher total SHQ gameplay duration and SHQ stress. We further predicted that more correct answers on PI levels would be associated with stronger VSWM and sense of direction as measured by D-Corsi and SBSOD, respectively. Moreover, we hypothesized that the number of reference memory errors (i.e., RM errors) and spatial working memory errors (i.e., SWM errors) on RAM levels would positively correlate with each other and negatively with D-Corsi and SBSOD scores. Additionally, we predicted that mental rotation and visuospatial processing, as measured by the MRT and the DOT, respectively, would positively correlate with each other and with VSWM, as measured by the D-Corsi. Our final hypothesis was that sense of direction and mapping tendency, as measured by the SBSOD and the NSQ, respectively, would positively correlate with each other. Table 1 below shows the 20 hypothesised relationships between performance on different tasks.

**Table 1. Hypothesised relationships between performance on various tasks.**

| | WF Inefficiency | PI Correct Answers | RAM RM Errors | RAM SWM Errors | SHQ Tutorial Duration | MRT Score | DOT Score | D-Corsi Score | SBSOD Score | NSQ Score | SHQ Stress Rating |
|---|---|---|---|---|---|---|---|---|---|---|---|
| WF Inefficiency | | | | | | | | | | | |
| PI Correct Answers | **-ve** | | | | | | | | | | |
| RAM RM Errors | | | | | | | | | | | |
| RAM SWM Errors | | | **+ve** | | | | | | | | |
| SHQ Tutorial Duration | **-ve** | | | | | | | | | | |
| MRT Score | -ve | | | | | | | | | | |
| DOT Score | -ve | | | | | +ve | | | | | |
| D-Corsi Score | -ve | +ve | -ve | -ve | | +ve | +ve | | | | |
| SBSOD Score | -ve | +ve | -ve | -ve | | | | | | | |
| NSQ Score | -ve | | | | | | | | +ve | | |
| SHQ Stress Rating | -ve | -ve | | | | | | | | | |

*Note.* The predictions shown are based on the existing literature and (in bold) on previous findings from SHQ (Coutrot et al. [8]; Hegarty et al. [18]; Hegarty et al. [38]; Lawton et al. [44]; West et al. [45]). Where there was no direct research on the relationship(s) between the specific measures used in this study, we relied on research on how various constructs assessed by the measures are associated with each other. For instance, our prediction on the relationship between wayfinding on SHQ and mapping tendency, as measured by the NSQ, is based on the finding that those who use a map-based strategy, relative to a non-map-based strategy, are generally considered to be better at navigation tasks [35]. Similarly, we hypothesised that sense of direction, as measured by the SBSOD, will be positively associated with navigation on SHQ given that better sense of direction is usually related to efficient navigation [46,47]. We predicted that gameplay stress will be negatively associated with navigation based on previous research that shows stress impairs wayfinding and path integration performance [48,49]. Lastly, the observation that individuals with high VSWM are faster and make fewer errors during wayfinding influenced our hypothesis about the relationship between SHQ wayfinding and VSWM as indicated by D-Corsi [19]. +ve = Positive correlation. -ve = Negative correlation. WF = Wayfinding. PI = Path Integration. RAM RM Errors = Radial Arm Maze Reference Memory Errors. RAM SWM Errors = Radial Arm Maze Spatial Working Memory Errors. MRT = Mental Rotation Test. DOT = Design Organization Test. D-Corsi = Digital Corsi Block Tapping Task. SBSOD = Santa Barbara Sense of Direction Scale. NSQ = Navigation Strategies Questionnaire. SHQ = Sea Hero Quest.

## 2. Methodology

### 2.1 Context of data collection

This article is an extension of broader research that focussed on the efficacy of SHQ in reducing the frequency of intrusive memories of analogue trauma using the trauma film paradigm. Reporting of findings related to SHQ and its effect on intrusive memories is beyond the scope of this paper. The design and analysis of this research were not pre-registered. The data for this study will be made publicly available.

### 2.2 Participants

Seventy-eight healthy English-speaking participants over the age of 18 years ($M$ = 20.33 years, $SD$ = 4.03 years) were recruited from University College London through its online subject pool SONA, and compensated in the form of course credits. Demographic information about the participants is summarised in Table 2. Ethical approval was obtained from the University College London Review Board (9807/004). All participants provided informed written consent.

### 2.3 sea hero quest tasks

**2.3.1 sea hero quest app (7, 8, 45).** SHQ is a VR navigation game for mobile and tablet devices in which participants are required to navigate a three-dimensional environment of

**Table 2. Demographics overview for participants in the study.**

| Age | | Gender | | Race | |
|---|---|---|---|---|---|
| 18–24 | 73 | Female | 58 | Asian | 45 |
| 25–30 | 4 | Male | 20 | White | 27 |
| 40–49 | 1 | | | Other | 4 |
| | | | | Black | 2 |

water bodies. Navigation abilities of participants are assessed through three types of levels: Wayfinding, PI and RAM (see Fig 1). To avoid a ceiling effect, participants played the somewhat "difficult" levels of SHQ. The difficulty of wayfinding levels was based on the number of goals in a particular level, and how far apart they were located from each other. Levels in which there were four or more goals that were located at a considerable distance from each other were considered difficult. For PI levels, difficulty was determined by the number of turns participants had to take to get from the starting point to the final destination. Levels in which there were at least four turns were selected. Based on the aforementioned criteria, we chose levels 16, 37, 32 and 43 for wayfinding to keep a broader selection of levels in terms of spatial characteristics of the environment. For PI, we selected levels 44, 49, 54, 59, 64, 69 and 74 that were later in the game and more challenging. For the purpose of this study, we only analysed

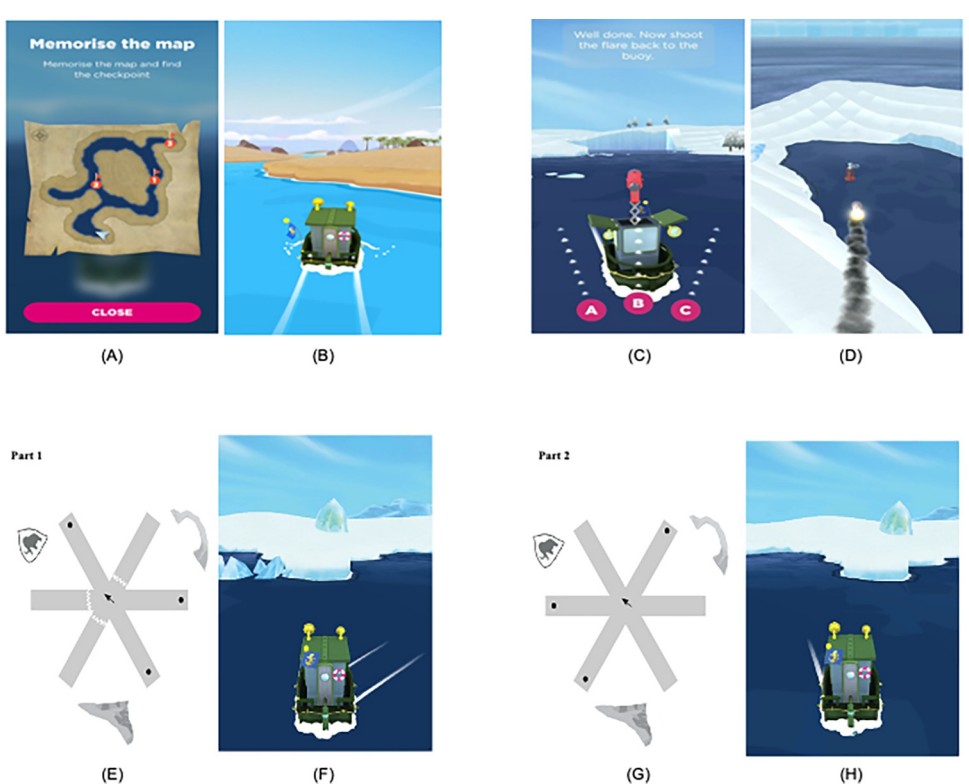

**Fig 1. Examples of the various types of levels in sea hero quest.** (A-B) Wayfinding Task: Participants memorise a map in the beginning of the task, and navigate to checkpoints in an ordered manner. (C-D): Path Integration (PI) Task: Participants navigate along a river to find a flare gun, and shoot the flare back to the starting point. (E-F) Radial Arm Maze Task (RAM) Part 1: Three of the six arms are blocked, and participants navigate to the three open arms to collect a star that pops out of the water in each of them. (G-H): Radial Arm Maze Task Part 2: All six arms are made available, and the participants are required to navigate to the three arms that were blocked during Part 1 to collect the remaining three stars.

levels that were played by all participants in the original trauma study. Accordingly, we considered levels 16 and 37 for wayfinding, and levels 44, 49, 54 and 74 for PI. Lastly, participants played all the RAM levels as there are only five such levels in SHQ.

In wayfinding levels, participants are required to navigate to checkpoints in an ordered manner based on a map presented to them before the game begins. Performance on wayfinding levels was operationalised as the average inefficiency across all the levels. Lower inefficiency values indicate better wayfinding performance, meaning that participants covered less distance to complete all the levels. To control for prior video gaming experience, we standardised wayfinding inefficiency by dividing it by the duration (in seconds) spent learning SHQ controls in the first two levels of the check-point (practice) task that required no spatial memory to solve. In PI levels, participants navigate along a river to find a flare gun, and shoot it back to the starting point by choosing one direction from three alternatives. Performance here was measured as the number of correct answers obtained at the end of all PI levels. Greater number of correct answers indicates better PI performance. The RAM levels are divided into two parts. In Part 1, three of the six arms are blocked, and participants visit the three free arms to collect a star from each of them. In Part 2, the goal of the participants is to visit the three arms that were blocked in Part 1, and collect the remaining three stars from them. Performance on RAM levels was operationalised as the number of reference memory (RM) and spatial working memory (SWM) "errors" made across all the RAM levels. RM errors refer to the number of visits to the arms already visited in Part 1 that needed to be avoided in Part 2. SWM errors refer to the number of visits made to arms in Part 2 that had already been visited during the second part itself. Fewer errors indicate better RAM performance.

**2.3.2 Sea hero quest stress rating.** Participants were asked to rate the stress they experienced while playing SHQ levels on a scale ranging from 0 ("*not at all*") to 10 ("*extremely*") once they completed gameplay.

## 2.4 Visuospatial ability measures

The visuospatial abilities of participants were measured using three tasks, which are as follows:

**2.4.1 Mental Rotation Test [23].** The MRT Form A consists of 12 stimuli, each of which is a two-dimensional image of a three-dimensional object drawn by a computer. We modified the MRT Form A to include three answer options and only one correct answer per question instead of four answer options and two correct answers, respectively. We also reduced the time limit from 4 minutes to 3 minutes to maintain adequate time pressure. Participants were presented the printed version, and they marked the answer option they thought was the rotated version of the target stimulus. Scores range from 0 to 12. Higher scores indicate better performance.

**2.4.2 Design Organization Test [25].** Participants completed the printed version of both Form A and Form B of the DOT in a counterbalanced manner. At the top of the page, there is a row of six squares that is numbered from 1 to 6, which serves as a code key for completing this task. There are nine square grids below this, each of which showcases a unique pattern composed of a specific combination of the numbered squares in the code key. Participants complete empty grids below these patterned grids using the corresponding numbered squares from the code key. As it is possible to get a ceiling effect within two minutes, participants were allotted one minute to complete this measure [50]. Scores range from 0 to 112. Higher scores indicate better performance.

**2.4.3 Digital Corsi [31].** In this task, participants observed a set of nine blocks on the computer screen. A number of the observed blocks lit up in a particular sequence, starting

with three blocks and increasing with every successful trial. Participants then repeated the sequence by clicking on the blocks in the order in which they lit up. The sum of the number of blocks they clicked in the correct order was computed as their total score, which could range from 0 to 150. Higher scores indicate better performance. It is important to note the digital version of the Corsi task is conceptually different from the regular manual version. Specifically, in the digital version, the absence of predictive finger movements results in slightly different processing and performance.

## 2.5 Navigation measures

Information about navigation preferences and strategies of participants was obtained using the measures listed below, which participants completed on a computer.

**2.5.1 Santa Barbara Sense of Direction Scale [38].**   The SBSOD was used to assess the perceived sense of direction of participants. The instrument consists of 15 items, which are measured on a seven-point scale ("*strongly agree*" to "*strongly disagree*"). Total scores range from 1 to 7. Higher scores indicate better perceived sense of direction. The SBSOD has been demonstrated to have high internal consistency ($\alpha = .88$) and test-retest reliability ($\alpha = .91$), which was assessed by administering the questionnaire 40 days apart. Construct validity was determined by significant correlations between SBSOD scores and wayfinding performance.

**2.5.2 Navigation Strategies Questionnaire [35].**   The NSQ consists of 14 items that evaluate the propensity for map-based navigation. The difference between the number of map-based answers and non-map-based answers is taken as a measure of mapping tendency. Total scores range from -14 to +14. Higher scores represent greater use of map-based strategy during navigation.

**2.5.3 Radial Arm Maze Navigation Strategy.**   After the completion of every RAM level, participants were presented with a multiple-choice question, which required them to indicate the type of navigation strategy they used to complete the level. Participants were asked, "*How did you navigate*? *(1) Counted from the start; (2) Used multiple landmarks; (3) Counted from the landmark.*" We recorded the most frequently used navigation strategy. Participants were categorised as having used a landmark strategy if they indicated either of the latter two options on at least three out of five RAM levels. The same was done for counting strategy if the first option was indicated on the majority of RAM levels. If participants skipped a question, and their dominant navigation strategy could not be determined (i.e., they had used counting strategy and landmark strategy on two levels each), their data were excluded from the analysis.

## 2.6 Procedure

The study was conducted over two sessions, which were held a week apart from each other. Before the first session, participants attempted the MRT to provide an initial measure of their visuospatial ability. In the first experimental session, participants played SHQ wayfinding and PI levels for six minutes each. Before starting SHQ gameplay, they completed practice levels as part of a brief tutorial to learn the controls of the game. At the end of the session, participants indicated their stress level during gameplay, and attempted the DOT. Upon their return a week later, participants played RAM levels, and indicated their navigation strategy for each level. They also performed the D-Corsi, and completed the SBSOD and the NSQ.

## 2.7 Power analysis

Based on the estimates from the prior work motivating the approach, the present study assumed the effect size of $d = 0.80$ [51]. Using the conversion calculations by Ruscio [52], we determined that $d = 0.80$ amounts to $r = 0.371$. Sample size calculations using these values

revealed that a minimum of 72 participants were required to achieve 90% power in order to detect a difference at the 5% significance level.

## 2.8 Data analysis

The data were analysed using IBM SPSS Statistics (Version 28) [53]. Extreme outliers were excluded from the analysis of the task in question, though their data for other tasks were retained if they were not outliers on them. Missing values were not inferred. Pearson correlation analysis was used to explore the relationship between SHQ and all the measures of visuo-spatial abilities, navigation and stress. Correlations for predicted relationships were evaluated at the $p < 0.05$ threshold (see Table 1), while all the other correlational analyses were Bonfer-roni-corrected for multiple comparisons, $p < 0.0009$ (55 comparisons) to control for Type I error. Independent samples $t$-tests were also performed to examine how performance between those who used landmark strategy or counting strategy most frequently on RAM levels differ on other measures. Finally, Pearson chi-square test of independence was used to examine the relationship between the propensity for map-based navigation (as indicated by the NSQ) and the type of navigation strategy used on RAM levels (used landmark strategy or counting strategy).

## 3. Results

We calculated the scores of participants across all the measures on which they were tested. The descriptive statistics of participant performance are summarised in Tables 3 and 4. Data from one participant was not recorded due to a technical error.

Independent 2-tailed $t$-tests revealed that participants who used landmark strategy on RAM levels made significantly fewer RM errors than those who used counting strategy. Such participants also had significantly lower wayfinding inefficiency and higher D-Corsi score on average than participants who employed counting strategy. Table 6 summarises statistics for variables that were significant.

Based on the results of Table 6, which are consistent with previous findings of West et al. [45], additional correlational analyses were conducted to check the association between both types of RAM errors and all SHQ and neuropsychological measures for RAM landmark strat-egy users only. We found a significant positive association between RAM RM errors and RAM

**Table 3. Descriptive statistics of participant performance across all measures.**

| Measure | Parameter | Overall | | | Females | | | Males | | |
|---|---|---|---|---|---|---|---|---|---|---|
| | | N | M | SD | n | M | SD | n | M | SD |
| SHQ | WF Inefficiency | 71 | 26.98 | 6.93 | 52 | 27.19 | 7.09 | 19 | 26.41 | 6.61 |
| | PI Correct Answers | 75 | 1.72 | 0.99 | 55 | 1.71 | 1.01 | 20 | 1.75 | 0.97 |
| | RAM RM Errors | 77 | 5.01 | 2.59 | 57 | 5.34 | 2.52 | 20 | 4.08 | 2.61 |
| | RAM SWM Errors | 77 | 0.94 | 1.13 | 57 | 1.07 | 1.19 | 20 | 0.58 | 0.85 |
| | SHQ Tutorial Duration | 74 | 46.52 | 6.45 | 55 | 47.36 | 6.81 | 19 | 44.08 | 4.61 |
| VSA | MRT Score | 78 | 7.37 | 3.10 | 58 | 7.07 | 3.25 | 20 | 8.25 | 2.51 |
| | DOT Score | 76 | 62.21 | 9.07 | 57 | 62.35 | 8.84 | 19 | 61.79 | 9.96 |
| | D-Corsi Score | 77 | 61.86 | 18.82 | 58 | 60.50 | 17.56 | 19 | 66.00 | 22.26 |
| Navigation Questionnaires | SBSOD Score | 77 | 4.10 | 0.77 | 57 | 4.02 | 0.74 | 20 | 4.32 | 0.84 |
| | NSQ Score | 77 | -1.83 | 5.03 | 57 | -2.44 | 4.80 | 20 | -0.10 | 5.38 |
| Gameplay Stress | SHQ Stress Rating | 77 | 3.96 | 2.33 | 57 | 4.44 | 2.39 | 20 | 2.60 | 1.50 |

*Note*. Sex differences are reported in the Appendix (S1 Appendix).

**Table 4. Strategy employed to navigate RAM levels.**

| Gender | n | Counting Strategy | Landmark Strategy |
|---|---|---|---|
| Female* | 58 | 12 (20.69%) | 45 (77.59%) |
| Male | 20 | 6 (30%) | 14 (70%) |
| Total | 78 | 18 | 59 |

Note.

\* = The dominant RAM navigation strategy of one female participant could not be determined due to missing data. Pearson correlation analysis revealed that all the measures of visuospatial abilities were associated with each other ($p < 0.05$, see Table 5). Similarly, responses on the two self-report measures of navigation were related, ($p < 0.05$, see Table 5). However, visuospatial and navigation measures were not correlated with each other ($p > 0.05$, see Table 5). Further, visuospatial abilities were only selectively associated with performance on SHQ wayfinding inefficiency. While all visuospatial abilities measures were associated with wayfinding inefficiency, they were not associated with SHQ tutorial duration. Only D-Corsi scores were associated with performance on PI levels. Neither measure of navigation was associated with SHQ wayfinding inefficiency. The significance of the association between gameplay stress and navigation preferences and strategies did not hold at the Bonferroni-corrected threshold ($p > 0.0009$). Gameplay stress was also found to be correlated with DOT, but again, the association was not significant at the Bonferroni-corrected threshold ($p > 0.0009$). The findings from correlation analyses are summarised in Table 5 and Fig 2. See the Appendix (S1 Appendix) for further supporting analysis of the large dataset reported in Coutrot et al. [12]. This shows the pattern of correlation between the training levels, PI levels and wayfinding performance varies across levels used in SHQ.

**Table 5. Correlation matrix for all participants.**

| | WF Inefficiency | PI Correct Answers | RAM RM Errors | RAM SWM Errors | SHQ Tutorial Duration | MRT Score | DOT Score | D-Corsi Score | SBSOD Score | NSQ Score |
|---|---|---|---|---|---|---|---|---|---|---|
| PI Correct Answers | 0.053 ($p = .664$) | | | | | | | | | |
| RAM RM Errors | 0.201 ($p = .093$) | 0.142 ($p = .224$) | | | | | | | | |
| RAM SWM Errors | 0.102 ($p = .398$) | -0.012 ($p = .920$) | 0.275 ($p = .016$) | | | | | | | |
| SHQ Tutorial Duration | -0.282 ($p = .018$) | -0.119 ($p = .320$) | -0.143 ($p = .223$) | 0.143 ($p = .223$) | | | | | | |
| MRT Score | -0.267 ($p = .024$) | 0.085 ($p = .470$) | -0.006 ($p = .956$) | 0.001 ($p = .990$) | -0.124 ($p = .292$) | | | | | |
| DOT Score | -0.304 ($p = .010$) | 0.124 ($p = .296$) | -0.060 ($p = .612$) | -0.013 ($p = .914$) | 0.016 ($p = .894$) | 0.269 ($p = .019$) | | | | |
| D-Corsi Score | -0.261 ($p = .029$) | 0.328 ($p = .004$) | -0.199 ($p = .085$) | -0.010 ($p = .930$) | -0.159 ($p = .178$) | 0.301 ($p = .008$) | 0.259 ($p = .025$) | | | |
| SBSOD Score | -0.054 ($p = .657$) | -0.060 ($p = .611$) | -0.024 ($p = .834$) | -0.039 ($p = .735$) | -0.159 ($p = .178$) | 0.039 ($p = .736$) | -0.023 ($p = .845$) | -0.032 ($p = .781$) | | |
| NSQ Score | -0.041 ($p = .734$) | 0.099 ($p = .400$) | -0.048 ($p = .682$) | -0.151 ($p = .193$) | -0.097 ($p = .416$) | 0.100 ($p = .387$) | 0.048 ($p = .683$) | 0.022 ($p = .851$) | 0.577 ($p < .001$) | |
| SHQ Stress Rating | -0.030 ($p = .804$) | -0.017 ($p = .885$) | 0.184 ($p = .110$) | 0.157 ($p = .173$) | -0.049 ($p = .679$) | 0.004 ($p = .972$) | 0.243 ($p = .036$) | 0.142 ($p = .221$) | -0.291 ($p = .011$) | -0.295 ($p = .010$) |

*Note.* The colour intensity of the cells indicates the strength of a correlation, red = negative associations and green = positive associations. Solid border indicates predicted relationships meeting criteria for significance ($p < 0.05$). Dotted border indicates unpredicted correlations that were significant at $p < 0.05$, but did not meet the Bonferroni-corrected threshold for significance ($p < 0.0009$). WF = Wayfinding. PI = Path Integration. RAM RM Errors = Radial Arm Maze Reference Memory Errors. RAM SWM Errors = Radial Arm Maze Spatial Working Memory Errors. MRT = Mental Rotation Test. DOT = Design Organization Test. D-Corsi = Digital Corsi Block Tapping Task. SBSOD = Santa Barbara Sense of Direction Scale. NSQ = Navigation Strategies Questionnaire. SHQ = Sea Hero Quest.

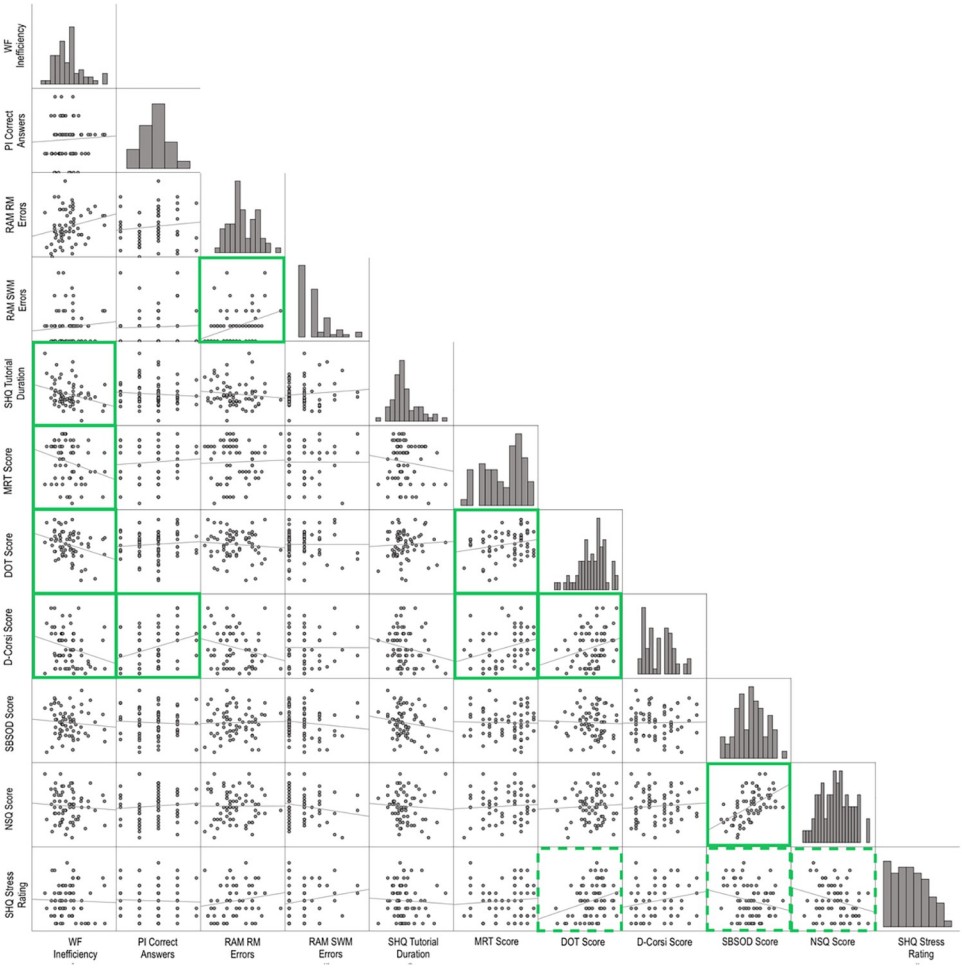

**Fig 2. Scatterplot matrix of relationships between 11 measures.** Solid green boxes indicate predicted significant relationships. Dotted green boxes indicate unpredicted relationships that were significant at $p < 0.05$, but did not meet the Bonferroni-corrected threshold for significance ($p < .0009$). WF = Wayfinding. PI = Path Integration. RAM RM Errors = Radial Arm Maze Reference Memory Errors. RAM SWM Errors = Radial Arm Maze Spatial Working Memory Errors. MRT = Mental Rotation Test. DOT = Design Organization Test. D-Corsi = Digital Corsi Block Tapping Task. SBSOD = Santa Barbara Sense of Direction Scale. NSQ = Navigation Strategies Questionnaire. SHQ = Sea Hero Quest. See Appendix (S1 Appendix) for further analysis.

SWM errors. Gameplay stress was also found to be correlated with RAM RM errors, but the association was not significant at the Bonferroni-corrected threshold ($p > 0.0009$). The findings are summarised in Table 7.

We did not find an association between mapping tendency as indicated by the NSQ and the navigation strategy used on RAM levels, $\chi2$ (1, N = 76) = 0.034, $p = 0.855$.

## 4. Discussion

We tested participants with virtual navigation tasks (wayfinding, PI and RAM) in the gaming app SHQ, visuospatial abilities (mental rotation, visuospatial processing and VSWM), and navigation strategies and preferences (sense of direction, mapping tendency and RAM navigation strategy) to better understand how these cognitive constructs relate to each other. The different constructs showed low levels of association, with negligible correlation among the three spatial navigation tasks on SHQ, and weak correlation between these and the self-ratings and

**Table 6. Metrics found to be significantly different for landmark strategy versus counting strategy on RAM levels.**

| Measure | Landmark Strategy | | Counting Strategy | | Statistics | | | |
|---|---|---|---|---|---|---|---|---|
| | M | SD | M | SD | t | df | p | d |
| WF Inefficiency | 25.712 | 6.502 | 31.004 | 6.887 | 2.888 | 69 | .005 | .802 |
| RAM RM Errors | 4.525 | 2.602 | 6.611 | 1.803 | 3.169 | 75 | .002 | .853 |
| D-Corsi Score* | 64.322 | 19.667 | 54.118 | 13.518 | -2.453 | 37.608 | .019 | -.551 |

*Note*. Non-significant metrics not shown.

* = Statistics reported for D-Corsi scores are not assuming equal variances as the Levene's Test was significant ($F = 4.995$, $p = 0.028$).

navigation strategies. We observed modest correlations between each of the three visuospatial abilities, and all of them with wayfinding, but not with other navigation tasks. We discuss what these results mean for understanding cognitive profiles of navigation ability.

Consistent with our predictions, we found a significant correlation between wayfinding performance and performance on visuospatial tasks of mental rotation, visuospatial processing and VSWM. To our knowledge, this is the first study to explore the relationship between visuospatial processing as measured by the DOT and navigation tasks. In all three cases, a low-to-moderate correlation is in line with previous studies examining the relationship between large-scale and small-scale spatial abilities [18,21]. The absence of strong correlations between these abilities indicates that while wayfinding and these constructs may have some overlap in the cognition required, they also have different demands. This is a pattern discussed in past research exploring the relation between small-scale spatial abilities and large-scale wayfinding ability using SHQ and real-world navigation tasks involving various measures such as accuracy, reaction time, distance travelled and number of errors [11]. These findings are useful in highlighting the potential utility of VR-based navigation tasks in capturing something that is distinct from object-based visuospatial tasks. One factor that may differentiate wayfinding

**Table 7. Correlations for RAM errors for RAM landmark strategy users only.**

| | RAM RM Errors | RAM SWM Errors |
|---|---|---|
| RAM SWM Errors | 0.352 ($p = 0.006$) | |
| WF Inefficiency | 0.131 ($p = 0.347$) | 0.117 ($p = 0.400$) |
| PI Correct Answers | 0.085 ($p = 0.529$) | 0.035 ($p = 0.794$) |
| SHQ Tutorial Duration | -0.070 ($p = 0.609$) | 0.112 ($p = 0.412$) |
| MRT Score | 0.012 ($p = 0.930$) | 0.087 ($p = 0.511$) |
| DOT Score | -0.047 ($p = 0.725$) | -0.040 ($p = 0.767$) |
| D-Corsi Score | -0.131 ($p = 0.323$) | 0.034 ($p = 0.801$) |
| SBSOD Score | -0.069 ($p = 0.609$) | -0.064 ($p = 0.632$) |
| NSQ Score | -0.130 ($p = 0.329$) | -0.155 ($p = 0.246$) |
| SHQ Stress Rating | 0.324 ($p = 0.012$) | 0.205 ($p = 0.120$) |

*Note*. The colour intensity of the cells indicates the strength of a correlation, red = negative associations and green = positive associations. Solid border indicates predicted relationships meeting criteria for significance ($p < 0.05$). Dotted border indicates unpredicted correlations that were significant at $p < 0.05$, but did not meet the Bonferroni-corrected threshold for significance ($p < .0009$). WF = Wayfinding. PI = Path Integration. RAM RM Errors = Radial Arm Maze Reference Memory Errors. RAM SWM = Radial Arm Maze Spatial Working Memory Errors. MRT = Mental Rotation Test. DOT = Design Organization Test. D-Corsi = Digital Corsi Block Tapping Task. SBSOD = Santa Barbara Sense of Direction Scale. NSQ = Navigation Strategies Questionnaire. SHQ = Sea Hero Quest.

from the other visuospatial tasks is the reliance on broader executive functions demands. Unlike object-based visuospatial tasks, wayfinding places specific demands on planning and inhibition [54,55]. In SHQ, this involves avoiding re-approaching visited checkpoints and planning optimal paths given the order of checkpoints indicated on the initially shown map.

Interestingly, the three tests of visuospatial ability have a moderate correlation with each other, indicating that they measure different but related cognitive aspects. Mental rotation may require flexible switching between motor simulation and analytic thinking, depending on the difficulty level [56]. Block tapping may involve control processes, visual working memory and visuospatial attention [57,58]. Visuospatial processing on the DOT may rely on both visuospatial abilities and problem-solving skills [59]. This may explain why visuospatial processing on the DOT has the strongest correlation with wayfinding performance, considering the link between wayfinding performance and executive functions.

We had predicted that performance on RAM levels and VSWM as measured by D-Corsi performance would be correlated because both require holding a set of locations in mind over many seconds i.e., they test spatial working memory. We found no evidence for this. This result is consistent with the view that 'spatial working memory' is not a unitary cognitive function, but depends on the context (e.g., 2D screen space versus VR-rendered 3D environment). Theoretically, it seems plausible that the D-Corsi is supported mainly via frontoparietal circuits, and that the RAM additionally draws on hippocampal circuits to support the representation of the large-scale environment [45,60]. The absence of a correlation between the RAM and other navigation tasks further indicates that different cognitive demands between the tasks are involved.

A surprising result was that neither sense of direction nor mapping tendency was correlated with navigation performance on any SHQ task. This stands in contrast with previous evidence of correlations between navigation behaviour and these constructs [35,38,61], but is similar to some other evidence [62]. It may be that participants need to physically move or use body-based cues to navigate in a space for a stronger association of navigation behaviours with standardised measures such as the SBSOD (e.g., Hegarty et al. [18]). Another possibility is that people tend to rate their navigation ability in relation to how often they might get lost. It is likely that participants use GPS-based systems to find their way, and might rarely find themselves in the situation simulated in the wayfinding task where a map is studied and must be committed to memory before navigation. When we recently sampled a large population, drawn from many nations and a range of ages, we found a consistent relationship between wayfinding performance in SHQ and self-rated navigation ability (9). Thus, the relationship between self-ratings and wayfinding performance is likely moderated by a wide range of factors (e.g., He & Hegarty [63], Hegarty et al. [18], van der Ham & Koutzmpi [64], van der Ham et al. [65]). Notably, while actual performance in the wayfinding task in SHQ was not correlated with sense of direction or mapping tendency, the stress ratings of playing SHQ were, even though they did not reach the stringent Bonferroni-corrected significance level. Thus, people who think they are good navigators, or those who tend to think using maps, might be less likely to find SHQ stressful. In future, it will be useful to explore more about spatial anxiety in daily life and SHQ performance to understand if they are linked more than the stress reported [44,66]. It is worth mentioning that the baseline stress and mood ratings of participants could not be recorded due to a technical error. Hence, it is possible that SHQ stress ratings were not an accurate measure of the stress caused by the navigation tasks; the stress may have simply been a result of being in a laboratory setting, or a prior event before the assessment.

We found that the SBSOD and the NSQ are moderately correlated with each other, which indicates that having a strong sense of direction is associated with using map-based strategy to navigate. To the authors' knowledge, the association between these two constructs has been

explored for the first time. This finding indicates that the two tests capture related dimensions of navigational profiles and attitudes, but not so overlapping as to be equivalent.

We found that those who used the landmark strategy on RAM levels performed significantly better on both RAM and wayfinding levels than those who used the counting strategy. This mirrors recent evidence from the large participant group of over 37,000 participants [45]. By combining the NSQ with the RAM tests for this study we reveal, for this population, that landmark-counting strategies appear to be orthogonal to the survey-route dimension captured by the NSQ. Notably, the lack of a correlation between wayfinding inefficiency and the RAM errors in this small lab sample matches our recent report of an absence of correlation between these in an online sample of over 37,000 participants [45]. The absence of a correlation of the PI measure and the other navigation tasks is notable in light of recent evidence that PI, rather than other spatial tests, might be particularly important for early detection of AD [67–69].

There are a number of limitations to our study that should be considered. First, our sample size was limited by the challenge of conducting in lab testing, which was central to our planned design. Relatedly, the sample was overwhelmingly composed of young and female participants, which could impact the external validity of the findings. Since previous research has shown that differences based on sex, age and culture exist in navigation performance and related behaviours, it would be helpful to use large samples to explore a broader range of ages, and compare gender and performance across nations in the future [7]. Nonetheless, our results elucidate the relationship among object-based spatial ability, navigation strategy and virtual navigation performance specifically for young women. Second, the visuospatial tasks we used here were adapted from clinically used tests aimed at detecting differences between groups, whereas our aim was to explore variation in the population. Thus, it may be useful to further develop such tests to help optimally explore individual differences [70]. Finally, as with all correlative approaches to assessing individual differences, the capacity of the tests used will depend partly on the variance with the data generated by the test in the group tested. In our current study, the PI task had less variability than others, and thus, this may have impacted our capacity to detect some relationship between PI and other measures. Nonetheless, we were able to observe predicted correlation between this test and the D-Corsi, indicating the variation present was still sufficient to detect some effects.

In conclusion, our study highlights that many tests and self-rating scales for navigation and visuospatial abilities can be highly non-overlapping, at least in a UK university student sample. Our results help further characterise what the different tests used in SHQ are related to. For example, we show that wayfinding has some overlap with object-based visuospatial tasks, whereas PI and RAM tasks have much less. The findings also point to the limitations of standardised tests of spatial cognition in capturing the nuances of navigation performance, particularly the distinction between small- and large-scale functioning. In future research it would be useful to probe a broader range of environments to understand how the complexity of the layouts (12) and landmark density [71,72] lead to higher or lower correlations with object-based visuospatial ability.

## Supporting information

**S1 Checklist. STROBE statement—checklist of items that should be included in reports of observational studies.**
(DOCX)

**S1 Appendix.**
(DOCX)

**S1 Dataset.**
(XLSX)

## Acknowledgments

We would like to extend a thank you for all the participants who volunteered in this research.

## Author Contributions

**Conceptualization:** Tanya Garg, Charlotte P. Malcolm, Victor Kovalets, Veronique D. Bohbot, Hugo J. Spiers.

**Data curation:** Tanya Garg.

**Formal analysis:** Tanya Garg.

**Funding acquisition:** Michael Hornberger, Hugo J. Spiers.

**Investigation:** Tanya Garg.

**Methodology:** Tanya Garg.

**Project administration:** Hugo J. Spiers.

**Resources:** Eva Zita Patai, Antoine Coutrot, Hugo J. Spiers.

**Supervision:** Mary Hegarty, Hugo J. Spiers.

**Writing – original draft:** Tanya Garg, Pablo Fernández Velasco.

**Writing – review & editing:** Charlotte P. Malcolm, Veronique D. Bohbot, Mary Hegarty, Hugo J. Spiers.

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
