## [Decision Letter · Decision Letter 0]

11 Jul 2023

PONE-D-23-12928The relationship between object-based spatial ability and virtual navigation performancePLOS ONE

Dear Dr. Garg,

Thank you for submitting your manuscript to PLOS ONE. After careful consideration, we feel that it has merit but does not fully meet PLOS ONE’s publication criteria as it currently stands. Therefore, we invite you to submit a revised version of the manuscript that addresses the points raised during the review process.

I would strongly urge the authors to give careful consideration to the reasoning, discussion, and conclusion presented in the manuscript, as well as the examination of the sample and its comparison to data from other articles published on SHQ and relevant literature. Extensive revisions and possible restructuring will be required, particularly in the Discussion and Conclusion sections of the document. Please read the reviewers' comments for further details. 

We look forward to receiving your revised manuscript.

Kind regards,

Amir-Homayoun Javadi, PhD

Academic Editor

PLOS ONE

Reviewers' comments:

Reviewer's Responses to Questions

**Comments to the Author**

1. Is the manuscript technically sound, and do the data support the conclusions?

Reviewer #1: Yes

Reviewer #2: Partly

2. Has the statistical analysis been performed appropriately and rigorously? 

Reviewer #1: Yes

Reviewer #2: Yes

3. Have the authors made all data underlying the findings in their manuscript fully available?

Reviewer #1: Yes

Reviewer #2: Yes

4. Is the manuscript presented in an intelligible fashion and written in standard English?

Reviewer #1: Yes

Reviewer #2: Yes

5. Review Comments to the Author

Reviewer #1: In their study „The relationship between object-based spatial ability and virtual navigation performance“, Garg et al. compare the performance in a mobile gaming app from 78 participants with established questionnaires or other tests for visuospatial abilities (mental rotation, visuospatial working memory, visuospatial processing, self-assessment of orientation). Surprisingly, only sparse strong correlations were found. The authors explain this finding by different subqualities being assessed in the individual tests.

The manuscript is overall well written and the methodology is sound. The mobile gaming app is well-established and its application in comparison with other test batteries can offer valuable insight into spatial abilities. However, my main concern is the small sample size, consisting of young, predominantly female participants. This factor needs further emphasis when discussing the results. One further aspect that would require major revision is the analysis of stress level during testing which is not compared to baseline stress or anxiety levels.

The manuscript fulfills the PLOS ONE publication criteria and, apart from the aforementioned concern, only requires minor revisions.

• Abstract: please provide information on the participants (N=78, 20 male, mean age 20.33yrs)

• Abstract: line 66 „disease monitoring in Alzheimer’s Disesase“: given how spatial impairment can occur in other neurological disorders outside of AD, consider rephrasing the sentence. A clinically applicable test for spatial impairment would benefit not only dementia-patients but other fields of neurology, too. (again in Introduction, line 95). E.g. Borel, L., et al. "Vestibular syndrome: a change in internal spatial representation." Neurophysiologie Clinique/Clinical Neurophysiology 38.6 (2008): 375-389., Mitchell, Peter, and Danielle Ropar. "Visuo‐spatial abilities in autism: A review." Infant and Child Development: An International Journal of Research and Practice 13.3 (2004): 185-198., …

• Introduction: line 95: „There are also negative effects of disorientation and its associated risks“: very vague, please explain what you mean by this.

• Introduction: line 109: „…can be partly predicted by gender inequality“ and line 111, „…peaks at seven hours of sleep…“: consider rephrasing these sentences, since right now the meaning remains unclear.

• The hypotheses in table 1 should rather be part of the introduction then their own table; in the current form, the table doesn’t add further value for the reader.

• Introduction, line 118: please indicate that a predominantly female, young cohort was tested.

• Methods 2.2, line 153: „no psychological disorder“: were any neurological disorders evident? Did any of the participants have a history of neurological diseases potentially affecting visuospatial abilities?

• Methods 2.4.1, line 219: how were the images presented? Printed, on screen, …?

• Methods 2.4.2: from the description alone, I had trouble understanding the DOT. Consider rephrasing the paragraph to make it clearer.

• Discusssion: line 406f: „it would seem obvious…“ Why would this seem obvious? Please define the discrepancy between the examples further. Is the hypothetical „new environment“ a virtual environment? Are tactile, vestibular, other sensory cues available? Rather than starting with such a vague statement, consider coming back towards the hypotheses from the introduction on what relationships and correlations you assumed based on the literature review. Alternatively, if available, start with studies on real life datasets where people with good performance in one spatial task tend to be good in others, too.

• Discussion line 425: Please specify that you refer to wayfinding in a mobile game rather than real-world navigation wayfinding , you might reference [15] again here while also please pointing out the methodical differences.

• Discussion line 453: „In the UK based sample…“ Do you think it’s an UK specific effect? If yes, please add information on why this UK cohort would especially rely on GPS to navigate. Otherweise please rephrase the segment.

• Discussion line 461: The whole analysis of stress level during task performance is not conclusive without information available on baseline stress/anxiety levels. Given the data collection period from 2018-2019, reevaluation of baseline stress is probably impossible – however, please discuss the absence of baseline stress level testing in its regard to the presented findings.

• Discussion line 500: „…at least in a UK university student sample.“ This is the main limitation of the study. This factor should be discussed in regards to all other results: are you observing general effects or is there bias due to the participant cohort?

Reviewer #2: I enjoyed reading the manuscript and was curious about the relevant topic of this work. I believe it concerns timely and meaningful data that should be published. The SHQ work is highly impactful and provides a strong and informative foundation for spatial cognition research. However, I also found the manuscript lacking in depth and theoretical detail in several place throughout the text. I would recommend some substantial additions and clarifications throughout the text before recommending publication of this work. Below I have listed my comments in order of appearance in the manuscript:

-Highlights: the assessment of self-ratings is mentioned without mention of outcome, and the outcomes of strategy are included without mention of its assessment. perhaps pick on of the two for the highlights section

-Abstract: The main comment that comes to mind in reading the manuscript is that terminology is used quite loosely. I would recommend a more clear and uniform terminology throughout. For instance, self-ratings are labelled as measures of 'confidence' in the abstract, which is a different construct which does not seem to be measured here. The phrase 'those good at wayfinding' is not clear in what group is referred to, those who were specifically good in the subtest of wayfinding of the SHQ? The manuscript would benefit from a thorough check of different terms used throughout. The field already suffers from a large variety of different terms, tasks, questionnaires, etc., it would be very helpful to provide clarity. I will include some examples below.

-Introduction: The first paragraph could be rewritten. The relevance of specifically including the tasks that the authors selected is not made clear, other than 'variation exists'. Variation exists for all cognitive abilities, so this does not provide any specific explanation. The introduction is very brief and lacks conviction for the task and sample selection in the current study. A related issue is that the tasks and questionnaires are referred to by their names, rather than the cognitive constructs involved. There is ample evidence suggesting that Corsi forward is more related to spatial attention than to spatial working memory for instance. PI is not provided in full the first time. There is no clear research question, the hypothesized correlations are quite elaborate, but not motivated other than in a brief table legend. I was a bit surprised to read that if no existing literature was available, correlations were based on 'general assumptions'. I would really be interested in learning the reasoning behind those assumptions.

-Methodology: the sample composition was very skewed in age group and gender, I wondered if this was included in the comparisons that were made with other samples (e.g. for the power analyses) and how the authors think this might affect their outcomes. The level selections were not motivated, there is mention of difficulty etc., but not why those particular levels of difficulty were worth selecting. The stress measurement seems rather limited, as this was a lab study, some individuals could have just been stressed about the lab setting, rather than the navigation task at hand, or some prior event right before the assessment, is there any measure of pre-test stress level available? If not, this seems important to include in the discussion. Both DOT and Corsi are not described based on the cognitive constructs measured. This adds to the confusion about the different terms used throughout the manuscript. Why was the Corsi backward not included?

-Discussion: I would highly recommend rewording the discussion based on the constructs measured rather than the names of all the different tasks, questionnaires, etc involved, at this point those names only have limited importance and a conceptual discussion would really help understanding the text.

-Conclusion: a main source of the limited overlap between measures would most likely be the highly specific task selection, in my opinion, which has not been motivated in this text. If the authors aimed to provide a comprehensive set of measures, this could be problematic, but from the introduction right now, this does not seem to be the case. It would be really helpful to flesh out the exact constructs across the measures, to understand the limited correlations. The future recommendations could be highly interesting but are not motivated here, it would be really helpful to provide a clear motivation for them.

6. PLOS authors have the option to publish the peer review history of their article (what does this mean?). If published, this will include your full peer review and any attached files.

Reviewer #1: No

Reviewer #2: No

---

## [Author Response · Author response to Decision Letter 0]

30 Sep 2023

EDITOR COMMENTS:

I would strongly urge the authors to give careful consideration to the reasoning, discussion, and conclusion presented in the manuscript, as well as the examination of the sample and its comparison to data from other articles published on SHQ and relevant literature. Extensive revisions and possible restructuring will be required, particularly in the Discussion and Conclusion sections of the document. 

>> Thank you for encouraging us to carefully consider these particular elements. We have substantially revised the manuscript in light of the reviews. 

REVIEWER 1 COMMENTS:

Overall:

In their study “The relationship between object-based spatial ability and virtual navigation performance“, Garg et al. compare the performance in a mobile gaming app from 78 participants with established questionnaires or other tests for visuospatial abilities (mental rotation, visuospatial working memory, visuospatial processing, self-assessment of orientation). Surprisingly, only sparse strong correlations were found. The authors explain this finding by different subqualities being assessed in the individual tests.

The manuscript is overall well written and the methodology is sound. The mobile gaming app is well-established and its application in comparison with other test batteries can offer valuable insight into spatial abilities. 

However, my main concern is the small sample size, consisting of young, predominantly female participants. This factor needs further emphasis when discussing the results.

>>We are grateful for the suggestion to discuss these issues further. They are indeed an important limitation of the study, and we believe adding discussion of them has strengthened the depth of the manuscript. Please refer to L567-573 in the discussion section.

>>“Relatedly, the sample was overwhelmingly composed of young and female participants, which could impact the external validity of the findings. Since previous research has shown that differences based on sex, age and culture exist in navigation performance and related behaviours, it would be helpful to use large samples to explore a broader range of ages, and compare gender and performance across nations in the future (7). Nonetheless, our results elucidate the relationship among object-based spatial ability, navigation strategy and virtual navigation performance specifically for young women.”

One further aspect that would require major revision is the analysis of stress level during testing which is not compared to baseline stress or anxiety levels.

>> Thank you for highlighting this. Unfortunately, due to a technical error at the time of data collection, the baseline stress and mood ratings of participants could not be recorded. We have now clarified as such in the discussion section, and have also discussed the implications of the absence of baseline stress levels. Please refer to L542-546 in the discussion section.

>> "It is worth mentioning that the baseline stress and mood ratings of participants could not be recorded due to a technical error. Hence, it is possible that SHQ stress ratings were not an accurate measure of the stress caused by the navigation tasks; the stress may have simply been a result of being in a laboratory setting, or a prior event before the assessment.”

The manuscript fulfills the PLOS ONE publication criteria and, apart from the aforementioned concern, only requires minor revisions.

>> Thank you for supporting minor revisions. 

Abstract:

1. Please provide information on the participants (N=78, 20 male, mean age 20.33yrs).

>> Thank you for pointing this out. We have now mentioned participant information in L68-L71 in the abstract section.

>> “Here, we report the use of an established mobile gaming app, Sea Hero Quest (SHQ), as a measure of navigation ability in a sample of young, predominantly female university students (N = 78; 20; female = 74.3%; mean age = 20.33 years).”

2. Line 66 “disease monitoring in Alzheimer’s Disease“: given how spatial impairment can occur in other neurological disorders outside of AD, consider rephrasing the sentence. A clinically applicable test for spatial impairment would benefit not only dementia-patients but other fields of neurology, too. (again, in Introduction, line 95). E.g., Borel, L., et al. "Vestibular syndrome: a change in internal spatial representation." Neurophysiologie Clinique/Clinical Neurophysiology 38.6 (2008): 375-389., Mitchell, Peter, and Danielle Ropar. "Visuospatial abilities in autism: A review." Infant and Child Development: An International Journal of Research and Practice 13.3 (2004): 185-198., …

>> Thank you for suggesting this. We have added multiple sclerosis, vestibular syndromes and autism as three more examples of disorders in L91-93 in the introduction section.

>> “Deficits in these competencies may constitute an early marker of conditions such as Alzheimer’s disease (AD), multiple sclerosis, vestibular syndrome and autism (1-4).”

Introduction:

1. Line 95: “There are also negative effects of disorientation and its associated risks“: very vague, please explain what you mean by this.

>> Thank you for raising this point. We have rephrased L93-94 in the introduction section to make our point clearer.

>> “Furthermore, navigation impairments like disorientation can affect one’s quality of life by causing distress and impediments in daily functioning (5, 6).”

2. Line 109: “…can be partly predicted by gender inequality“ and line 111, “…peaks at seven hours of sleep…“: consider rephrasing these sentences, since right now the meaning remains unclear.

>> Thank you for letting us know. We have rephrased L112-116 to make this information easier to understand.

>> “Studies using SHQ have revealed that gender differences in navigation ability for a country can be partially attributed to gender inequality (8). They have also found that individuals are more adept at navigating environments that are topologically similar to those in which they were raised (12). Such studies have also shown that performance is best for older participants who report sleeping 7 hours per night (13).”

3. The hypotheses in table 1 should rather be part of the introduction then their own table; in the current form, the table doesn’t add further value for the reader.

>> Thank you for suggesting this. We have added another paragraph at the end of the introduction section (L171-188) to make the predictions in Table 1 part of the text. However, we have also retained Table 1 as we feel it complements the newly added paragraph and makes it easier to visualise all of the 20 predictions in our study.

>> “In this study, we explore how visuospatial abilities, navigation strategy and gameplay stress relate to performance on SHQ and to each other. Based on the literature (e.g., Wolbers & Hegarty (17)), we made 20 predictions. We expected wayfinding to correlate with other measures due to its diverse demands such as perception, memory, decision-making, etc. Specifically, we predicted that longer duration to complete wayfinding levels (i.e., wayfinding inefficiency) would be associated with lower mental rotation, visuospatial processing, VSWM, sense of direction and mapping tendency as measured by the MRT, DOT, D-Corsi, SBSOD and NSQ, respectively. Greater wayfinding inefficiency would also be related to higher total SHQ gameplay duration and SHQ stress. We further predicted that more correct answers on PI levels would be associated with stronger VSWM and sense of direction as measured by D-Corsi and SBSOD, respectively. Moreover, we hypothesized that the number of reference memory errors (i.e., RM errors) and spatial working memory errors (i.e., SWM errors) on RAM levels would positively correlate with each other and negatively with D-Corsi and SBSOD scores. Additionally, we predicted that mental rotation and visuospatial processing, as measured by the MRT and the DOT, respectively, would positively correlate with each other and with VSWM, as measured by the D-Corsi. Our final hypothesis was that sense of direction and mapping tendency, as measured by the SBSOD and the NSQ, respectively, would positively correlate with each other. Table 1 below shows the 20 hypothesised relationships between performance on different tasks.”

4. Line 118: please indicate that a predominantly female, young cohort was tested.

>> Thank you for raising this. We have now edited the manuscript to specifically mention this in L68-71 in the abstract, L160-162 in the introduction and L542-546 (please see our first response under the “overall” sub-header for Reviewer 1 Comments) in the discussion section.

>> L68-71: “Here, we report the use of an established mobile gaming app, Sea Hero Quest (SHQ), as a measure of navigation ability in a sample of young, predominantly female university students (N = 78; 20; female = 74.3%; mean age = 20.33 years).”

>> L160-162: “We test the navigation abilities of a predominantly female sample of university students on three tasks in SHQ: wayfinding, path integration (PI), and the radial arm maze (RAM) test of spatial memory.”

Methods:

1. Line 153: „no psychological disorder“: were any neurological disorders evident? Did any of the participants have a history of neurological diseases potentially affecting visuospatial abilities?

>> Participants were only asked about a history of psychological disorders. Nonetheless, we did not observe any evident neurological disorders, and participants did not report any difficulties preventing them from performing the tasks well.

2. Methods 2.4.1, line 219: how were the images presented? Printed, on screen, …?

>> Thank you for pointing this out. We have now specified the modality of test administration for all the questionnaires and tasks used in the study. Please see pages 14-15.

>> L282-283: “Participants were presented the printed version, and they marked the answer option they thought was the rotated version of the target stimulus.”

>> L287-288: “Participants completed the printed version of both Form A and Form B of the DOT in a counterbalanced manner.”

>> L305-306: “Information about navigation preferences and strategies of participants was obtained using the measures listed below, which participants completed on a computer.”

3. Methods 2.4.2: from the description alone, I had trouble understanding the DOT. Consider rephrasing the paragraph to make it clearer.

>> Thank you for flagging this to us. We have rephrased the paragraph about DOT to make it easier to understand. Please see L287-294 in the methodology section.

>> “Participants completed the printed version of both Form A and Form B of the DOT in a counterbalanced manner. At the top of the page, there is a row of six squares that is numbered from 1 to 6, which serves as a code key for completing this task. There are nine square grids below this, each of which showcases a unique pattern composed of a specific combination of the numbered squares in the code key. Participants complete empty grids below these patterned grids using the corresponding numbered squares from the code key. As it is possible to get a ceiling effect within two minutes, participants were allotted one minute to complete this measure (50). Scores range from 0 to 112. Higher scores indicate better performance.”

Discussion:

1. Line 406f: „it would seem obvious…“ Why would this seem obvious? Please define the discrepancy between the examples further. Is the hypothetical „new environment“ a virtual environment? Are tactile, vestibular, other sensory cues available? Rather than starting with such a vague statement, consider coming back towards the hypotheses from the introduction on what relationships and correlations you assumed based on the literature review. Alternatively, if available, start with studies on real life datasets where people with good performance in one spatial task tend to be good in others, too.

>> Apologies for the confusion, and thank you for your suggestions. We have now modified the first paragraph of the discussion section (L476-484) to tie it more closely to our hypotheses from the introduction section based on a review of the literature.

>> “We tested participants with virtual navigation tasks (wayfinding, PI and RAM) in the gaming app SHQ, visuospatial abilities (mental rotation, visuospatial processing and VSWM), and navigation strategies and preferences (sense of direction, mapping tendency and RAM navigation strategy) to better understand how these cognitive constructs relate to each other. The different constructs showed low levels of association, with negligible correlation among the three spatial navigation tasks on SHQ, and weak correlation between these and the self-ratings and navigation strategies. We observed modest correlations between each of the three visuospatial abilities, and all of them with wayfinding, but not with other navigation tasks. We discuss what these results mean for understanding cognitive profiles of navigation ability.”

2. Line 425: Please specify that you refer to wayfinding in a mobile game rather than real-world navigation wayfinding , you might reference [15] again here while also please pointing out the methodical differences.

>> Thank you for spotting this. We have clarified the use of Sea Hero Quest in L493-496 of the discussion section.

>> “This is a pattern discussed in past research exploring the relation between small-scale spatial abilities and large-scale wayfinding ability using SHQ and real-world navigation tasks involving various measures such as accuracy, reaction time, distance travelled and number of errors (11).”

3. Line 453: „In the UK based sample…“ Do you think it’s an UK specific effect? If yes, please add information on why this UK cohort would especially rely on GPS to navigate. Otherwise please rephrase the segment.

>> Thank you for pointing this out. We have rephrased this sentence (529-532) in the discussion section

>> “It is likely that participants use GPS-based systems to find their way, and might rarely find themselves in the situation simulated in the wayfinding task where a map is studied and must be committed to memory before navigation.”

4. Line 461: The whole analysis of stress level during task performance is not conclusive without information available on baseline stress/anxiety levels. Given the data collection period from 2018-2019, re-evaluation of baseline stress is probably impossible – however, please discuss the absence of baseline stress level testing in its regard to the presented findings.

>> Please refer to our second response under the “overall” sub-header for Reviewer 1 Comments.

5. Line 500: „…at least in a UK university student sample.“ This is the main limitation of the study. This factor should be discussed in regards to all other results: are you observing general effects or is there bias due to the participant cohort?

>> Please refer to our first response under the “overall” sub-header for Reviewer 1 Comments.

REVIEWER 2 COMMENTS AND RESPONSES

Overall:

I enjoyed reading the manuscript and was curious about the relevant topic of this work. I believe it concerns timely and meaningful data that should be published. The SHQ work is highly impactful and provides a strong and informative foundation for spatial cognition research. However, I also found the manuscript lacking in depth and theoretical detail in several places throughout the text. I would recommend some substantial additions and clarifications throughout the text before recommending publication of this work. Below I have listed my comments in order of appearance in the manuscript:

>> Thank you for your detailed feedback. We have substantially revised the manuscript in light of all the recommendations. 

Highlights:

The assessment of self-ratings is mentioned without mention of outcome, and the outcomes of strategy are included without mention of its assessment. perhaps pick one of the two for the highlights section

>> Thank you for pointing this out. We have edited the first bullet in the highlights section (L37-39).

>> “Three navigation test

---

## [Decision Letter · Decision Letter 1]

24 Oct 2023

PONE-D-23-12928R1The relationship between object-based spatial ability and virtual navigation performancePLOS ONE

Dear Dr. Garg,

Thank you for submitting your manuscript to PLOS ONE. After careful consideration, we feel that it has merit but does not fully meet PLOS ONE’s publication criteria as it currently stands. Therefore, we invite you to submit a revised version of the manuscript that addresses the points raised during the review process.

There are only a few very minor comments by the reviewers. I thought it would be best to amend the document accordingly before sending it to publication. I will not send the document to be reviewed again. 

We look forward to receiving your revised manuscript.

Kind regards,

Amir-Homayoun Javadi, PhD

Academic Editor

PLOS ONE

Journal Requirements:

Reviewers' comments:

Reviewer's Responses to Questions

**Comments to the Author**

1. If the authors have adequately addressed your comments raised in a previous round of review and you feel that this manuscript is now acceptable for publication, you may indicate that here to bypass the “Comments to the Author” section, enter your conflict of interest statement in the “Confidential to Editor” section, and submit your "Accept" recommendation.

Reviewer #1: (No Response)

Reviewer #2: (No Response)

2. Is the manuscript technically sound, and do the data support the conclusions?

Reviewer #1: Yes

Reviewer #2: Yes

3. Has the statistical analysis been performed appropriately and rigorously? 

Reviewer #1: Yes

Reviewer #2: Yes

4. Have the authors made all data underlying the findings in their manuscript fully available?

Reviewer #1: Yes

Reviewer #2: Yes

5. Is the manuscript presented in an intelligible fashion and written in standard English?

Reviewer #1: Yes

Reviewer #2: Yes

6. Review Comments to the Author

Reviewer #1: The revision addressed most of the concerns by both reviewers. Some aspects that would have improved the manuscript have now been mentioned as relevant limitations in the discussion section; sadly, no corrections for baseline stress levels were performed due to data inavailability. The method section is now improved and better readable, some relevant information on testing procedures has been added. Overall, the readability and scope of the manuscript has been improved while the basic limitation of a small, predominatly young and female sample size remains, but is diswcussed now.

All in all, I still have some minor comments on the text which should be corrected before publication, but they don't require another round of peer review.

Line 91-93: spatial competency is NOT an early marker of autism or vestibular syndromeS, but a possible symptom/another affected domain. Instead, the lines could read "Deficits in these competencies may constitute an early marker of degenerative conditions such as Alzheimer’s disease (AD); furthermore, spatial abilities are impaired in various neurological disorders or conditions, such as multiple sclerosis, vestibular syndromes and autism (1-4)."

116 ...sleeping 7 hrs per night" compared to what other group?

Thanks to the authors for addressing the reviewers comments.

Reviewer #2: The revision has been thorough, I appreciate the authors' efforts to incorporate all issues raised. I have two minor points that I would recommend to address and one comment I would like to share:

-abstract: In the final sentence, the authors indicate 'support', which would imply a causal relationship between the variables mentioned, I would suggest to avoid such implication as this was not tested.

-conclusion: This is much clearer now. I think the neuropsychological literature concerning navigation ability could be used here to further support this conclusion. It is notably difficult to try to capture navigation performance with standardized testing materials concerning spatial cognition, your results highlight the mostly likely reasons for this. Most likely, the distinction between small and large scale spatial functioning (only partially overlapping), is the central cue here. Perhaps the authors could incorporate this connection (to limitations of standardized testing materials, and small vs large scale functioning) to further strengthen their position

Final thought: the digital version of the Corsi is conceptually different from the regular manual corsi task. Here, the predictive finger movements are absent, which result in slightly different processing and performance.

7. PLOS authors have the option to publish the peer review history of their article (what does this mean?). If published, this will include your full peer review and any attached files.

Reviewer #1: No

Reviewer #2: No

---

## [Author Response · Author response to Decision Letter 1]

12 Jan 2024

REPLY TO REVIEWS:

We are grateful to have the opportunity to reply to the reviews and re-submit.

Below we quote comments from the Editor and Reviewers in italics, our response in blue font and quotes from our updated manuscript in red font (as in the red font in the re-submitted manuscript).

EDITOR COMMENTS:

There are only a few very minor comments by the reviewers. I thought it would be best to amend the document accordingly before sending it to publication. I will not send the document to be reviewed again. 

RESPONSE: Thank you for giving us the opportunity to revise the manuscript. We have addressed the minor comments made by the reviewers.

REVIEWER 1 COMMENTS:

Overall:

The revision addressed most of the concerns by both reviewers. Some aspects that would have improved the manuscript have now been mentioned as relevant limitations in the discussion section; sadly, no corrections for baseline stress levels were performed due to data unavailability. The method section is now improved and better readable, some relevant information on testing procedures has been added. Overall, the readability and scope of the manuscript has been improved while the basic limitation of a small, predominantly young and female sample size remains, but is discussed now.

All in all, I still have some minor comments on the text which should be corrected before publication, but they don't require another round of peer review.

RESPONSE: Thank you for your comments!

Introduction:

Line 91-93: spatial competency is NOT an early marker of autism or vestibular syndromes, but a possible symptom/another affected domain. Instead, the lines could read "Deficits in these competencies may constitute an early marker of degenerative conditions such as Alzheimer’s disease (AD); furthermore, spatial abilities are impaired in various neurological disorders or conditions, such as multiple sclerosis, vestibular syndromes and autism (1-4)."

RESPONSE: Thank you for pointing out this nuance. We have included the suggested phrasing in the manuscript (pg. 4, L91-94).

“Deficits in these competencies may constitute an early marker of degenerative conditions such as Alzheimer’s disease (AD); furthermore, spatial abilities are impaired in various neurological disorders or conditions, such as multiple sclerosis, vestibular syndromes and autism (1-4).”

116 ...sleeping 7 hrs per night" compared to what other group?

RESPONSE: We have now included information about the comparison groups (pg. 6, L116-117).

“Such studies have also shown that performance is best for older participants who report sleeping 7 hours per night compared to those reporting more or less sleep (13).”

Thanks to the authors for addressing the reviewers comments.

RESPONSE: We thank you for your thoughtful comments!

REVIEWER 2 COMMENTS:

Overall:

The revision has been thorough, I appreciate the authors' efforts to incorporate all issues raised. I have two minor points that I would recommend to address and one comment I would like to share:

Abstract:

In the final sentence, the authors indicate 'support', which would imply a causal relationship between the variables mentioned, I would suggest to avoid such implication as this was not tested.

RESPONSE: Thank you for raising this point. We have now rephrased this sentence (pg. 4, L80-81).

“These findings help clarify the associations between different abilities involved in spatial navigation.”

Conclusion:

This is much clearer now. I think the neuropsychological literature concerning navigation ability could be used here to further support this conclusion. It is notably difficult to try to capture navigation performance with standardized testing materials concerning spatial cognition, your results highlight the mostly likely reasons for this. Most likely, the distinction between small and large scale spatial functioning (only partially overlapping), is the central cue here. Perhaps the authors could incorporate this connection (to limitations of standardized testing materials, and small vs large scale functioning) to further strengthen their position.

RESPONSE: Thank you for this suggestion. We have added a sentence about this in the conclusion (pg. 30, L590-592).

“The findings also point to the limitations of standardised tests of spatial cognition in capturing the nuances of navigation performance, particularly the distinction between small- and large-scale functioning.”

Final Thoughts: 

The digital version of the Corsi is conceptually different from the regular manual Corsi task. Here, the predictive finger movements are absent, which result in slightly different processing and performance.

RESPONSE: Thank you for sharing your thoughts with us. We have added the following clarification under the “Digital Corsi” sub-header in the methodology section (pg. 15, L303-305).

“It is important to note the digital version of the Corsi task is conceptually different from the regular manual version. Specifically, in the digital version, the absence of predictive finger movements results in slightly different processing and performance.”

---

## [Editor Report · Decision Letter 2]

20 Jan 2024

The relationship between object-based spatial ability and virtual navigation performance

PONE-D-23-12928R2

Dear Dr. Garg,

We’re pleased to inform you that your manuscript has been judged scientifically suitable for publication and will be formally accepted for publication once it meets all outstanding technical requirements.

Kind regards,

Amir-Homayoun Javadi, PhD

Academic Editor

PLOS ONE
---

## [Editor Report · Acceptance letter]

26 Apr 2024

PONE-D-23-12928R2 

PLOS ONE

Dear Dr. Garg, 

I'm pleased to inform you that your manuscript has been deemed suitable for publication in PLOS ONE. Congratulations! Your manuscript is now being handed over to our production team.

Kind regards, 

on behalf of

Dr. Amir-Homayoun Javadi 

Academic Editor

PLOS ONE